# Methodological Aspects of Induced Sputum

**Silvano Dragonieri** [1] , **Andras Bikov** [2,3,*] , **Alessandro Capuano** [1] , **Simone Scarlata** [4] **and Giovanna Elisiana Carpagnano** [1]

1   Department of Respiratory Diseases, University of Bari, 70124 Bari, Italy; silvano.dragonieri@uniba.it (S.D.); alecapuano92@gmail.com (A.C.); elisiana.carpagnano@uniba.it (G.E.C.)
2   Manchester Academic Health Science Centre, Wythenshawe Hospital, Manchester University NHS Foundation Trust, Manchester M13 9WL, UK
3   Division of Infection, Immunity and Respiratory Medicine, Faculty of Biology, Medicine and Health, The University of Manchester, Manchester M13 9PT, UK
4   Department of Internal Medicine, Unit of Respiratory Pathophysiology and Thoracic Endoscopy, Bio-Medical Campus, 00128 Rome, Italy; s.scarlata@policlinicocampus.it
*   Correspondence: andras.bikov@mft.nhs.uk

**Highlights:**

   **What are the main findings?**

-   Induced sputum is a valuable non-invasive and cost-effective method for obtaining lower airway secretions, especially in patients who cannot produce sputum naturally.
-   Despite some technical demands and limitations, induced sputum offers significant advantages as a preferred alternative for large-scale and repeated airway sampling in respiratory conditions like asthma and COPD.

   **What is the implication of the main finding?**

-   The use of induced sputum provides researchers and clinicians with valuable insights into the cellular and biochemical components of airway secretions, allowing for a better understanding of airway inflammation, immune responses, and treatment efficacy in several respiratory conditions.
-   This allows for large-scale and repeated airway sampling, enabling researchers and clinicians to conduct comprehensive studies and monitor treatment responses over time, ultimately contributing to better patient care and improved outcomes.

**Abstract:** We aimed to conduct a state-of-the-art review of the current literature and offer further insights into the methodological aspects concerning induced sputum. The increasing popularity of sputum induction as a non-invasive and cost-effective method for obtaining lower airway secretions from patients who cannot produce sputum naturally has led to extensive research and applications in respiratory conditions like asthma and COPD. This technique allows for analysis of the cellular and biochemical components of the sputum to take place, providing insights into airway inflammation, immune cells, and help in predicting treatment response. Furthermore, induced sputum enables various analyses, including microRNA and gene expression studies and immunophenotyping. The procedure is generally safe and well tolerated, even in patients with airflow limitations; however, monitoring lung function is essential, especially in those with airway hyperresponsiveness. Optimal saline solution concentration and inhalation duration have been investigated, recommending a 15–20 min induction with hypertonic saline. Expectoration involves coughing at the end of each inhalation time. Careful handling during sputum processing is necessary for obtaining accurate results in cell cytology, immunocytochemistry, and in situ hybridization. Overall, induced sputum offers significant advantages as a preferred alternative for large-scale and repeated airway sampling, despite some technical demands and limitations.

**Keywords:** induced sputum; sputum; airway inflammation; non-invasive biomarkers

## 1. Introduction

The main purpose of sputum induction is to obtain a sufficient sample of lower airway secretions from patients who cannot produce sputum naturally (Figure 1) [1]. The induction process involves inhaling nebulized saline solution (either isotonic or hypertonic) over different time intervals and subsequently expectorating the secretions into a Petri dish (Figure 2) [1].

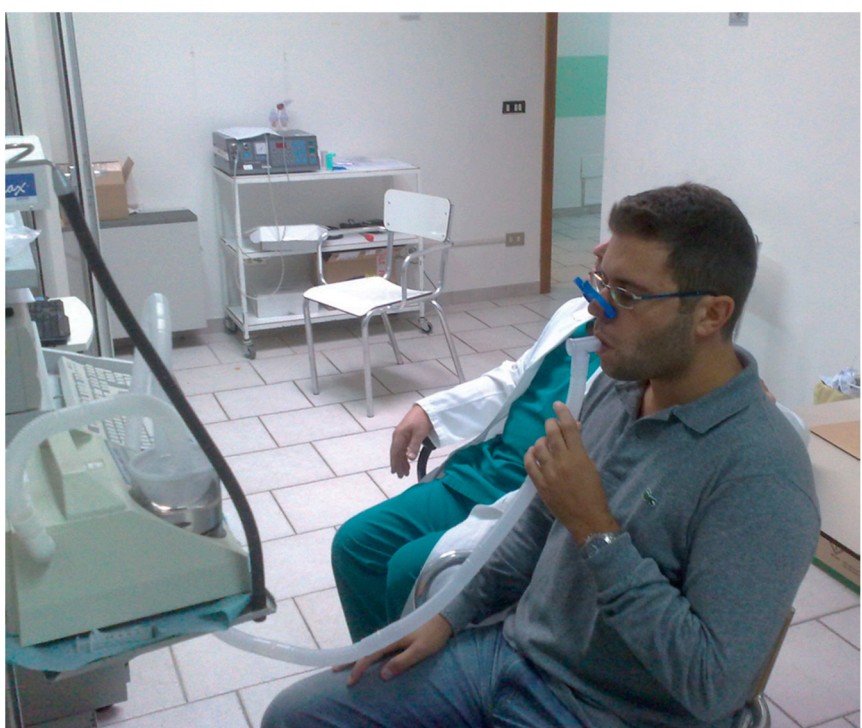

**Figure 1.** Sputum induction on a healthy volunteer (author S.D.).

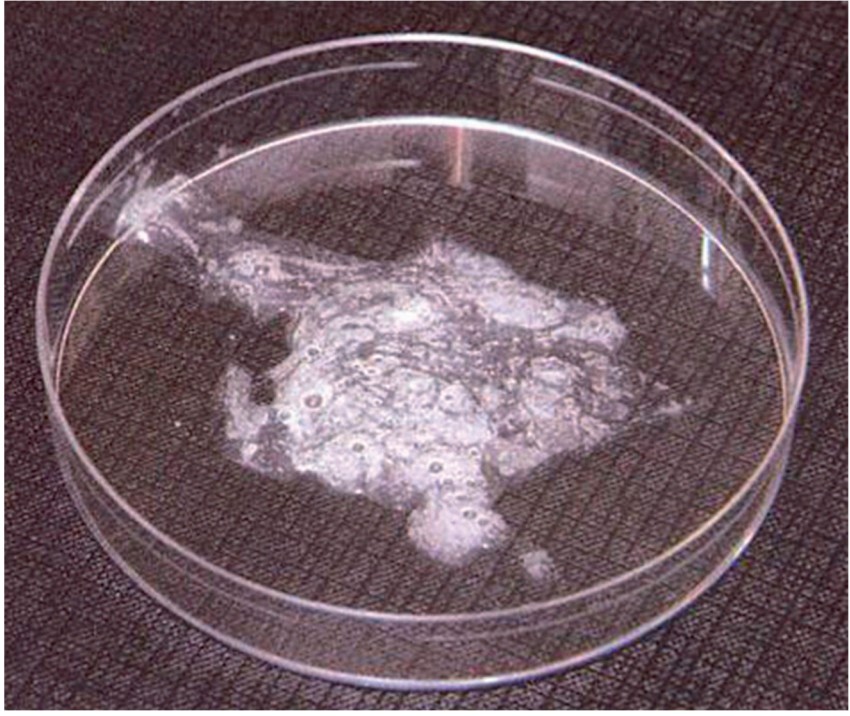

**Figure 2.** Sample of induced sputum in a Petri dish.

Over the past decades, there has been a significant increase in the utilization of sputum induction through inhaling hypertonic saline solution to examine the cellular and biochemical composition of the airways [2,3]. This induced sputum technique offers a relatively non-invasive way to sample airway content and has become more popular due to its cost-effectiveness and simplicity, as it does not require complex equipment [3]. It allows the collection of adequate samples of lower airway secretions, even from patients who cannot produce sputum spontaneously, enabling the investigation of airway inflammation in various respiratory conditions [3]. Firstly, it can enhance our understanding of immune cells and mechanisms involved in respiratory diseases, such as asthma and COPD, by investigating airway inflammation in large groups of patients [4]. Secondly, induced sputum can help predict treatment response, with high sputum eosinophil levels being indicative of corticosteroid responsiveness [5]. Tailoring corticosteroid doses based on sputum eosinophil percentages has shown better results in reducing exacerbations compared to standard guidelines [6]. Thirdly, sputum analysis can aid in developing targeted therapies, as abnormal sputum neutrophil counts have led to the development of antineutrophilic treatments [7]. Lastly, the technique may play a role in diagnosing non-asthmatic eosinophilic bronchitis based on the presence of sputum eosinophilia [8].

Additionally, induced sputum allows for various other analyses to take place, including the study of sputum supernatant and sputum cells. Examples of sputum supernatant analysis include the examination of mediators and chemotactic activity for eosinophils. From sputum cells, RNA extraction allows for microRNA and gene expression analyses, and flow cytometry permits immunophenotyping and cell sorting [9,10]. Furthermore, sputum cells can be cultured to measure mediator production in vitro. However, the mediator content differs from that found in sputum supernatant due to the influences from airway resident cells and plasma exudation [9,10].

Numerous studies have demonstrated that sputum induction is a reproducible, sensitive, and valid method [11,12]. Since the standardized method was first described [13], there has been a remarkable surge of methodological studies for expanding our understanding of critical aspects, such as the optimal method for inducing sputum and the techniques for its homogenization and processing [14].

While the induced sputum technique offers valuable insights into respiratory health, it is not without its limitations. This approach necessitates careful medical oversight, comprehensive patient instructions, and the willing cooperation and health status of the patient. Although there have been reports of successful use in children older than six years, applying this method to younger children can pose significant challenges [14]. Furthermore, the utilization of this technique is currently confined to research facilities and specialized centers due to its technical intricacies and the requirement for well-trained personnel [4].

The effectiveness of sputum induction and subsequent analysis hovers at approximately 80%, indicating that it may not be uniformly successful for all individuals [4]. Therefore, when comparing different patient groups, it is essential to exercise caution and consider variables such as age, gender, and smoking history as they can influence the outcomes and interpretation of induced sputum results [14].

Based on the above, we aimed to conduct a comprehensive review of the current literature and offer further insights into the methodological aspects concerning induced sputum.

## 2. Safety Considerations

Sputum induction is generally considered to be a safe procedure [14]. No fatalities or hospitalizations have been reported in subjects undergoing sputum induction. The potential adverse effect of bronchoconstriction caused by inhaling hypertonic saline solution can be rapidly reversed by administering inhaled short-acting β-2 agonists. Sputum induction is well-tolerated in patients with mild to moderate airflow limitations, such as those with asthma and COPD [15–19]. With the use of a modified procedure, it can also be safely performed in patients with moderate to severe airflow limitation [20], and even in patients experiencing an ongoing exacerbation of COPD [21]. Although there may be

considerable decreases in FEV1 during the induction procedure, patients generally tolerate this well. It has also been demonstrated that sputum induction can be safely conducted in children with mild to moderate asthma [22], as well as in individuals experiencing exercise-induced bronchoconstriction [23]. However, it is important not to overlook the risk of airway constriction in subjects with airway hyperresponsiveness (AHR). To mitigate this risk, pretreatment with salbutamol, a short-acting β-2 agonist, is recommended. Some authors suggest using a higher dose of 400 μg instead of 200 μg to prevent breakthrough bronchospasm, as the higher dose does not compromise the effectiveness of any additional required doses during the procedure [14]. To ensure safety during the induction, it is essential to monitor lung function using a spirometer as it provides higher sensitivity compared to a peak flow meter. Spirometry should be performed both before the induction procedure to establish baseline FEV1 and 10 min after the administration of preinduction inhaled β-2 agonist. Additionally, pulmonary function should be measured after 1 min of nebulization, as some individuals may be highly sensitive to hypertonic saline. After each inhalation interval, spirometry should be conducted. The procedure must be immediately and irretrievably stopped if FEV1 falls to more than 20% of the post-bronchodilator baseline or if the patient experiences breathlessness or wheezing. In such cases, an additional dose of β-2 agonists should be administered. Induction can resume only after FEV1 has returned to 95% of the post-bronchodilator baseline. At the end of the last inhalation procedure, if FEV1 has decreased to more than 10% of the post-bronchodilator baseline, an additional dose of β-2 agonist is recommended. Patients should be continuously monitored until their FEV1 reaches at least 95% of the baseline value [24].

## 3. Saline Solution Concentration and Inhalation Duration

Researchers have utilized various concentrations of saline solution, ranging from 0.9% to 7%, for sputum induction [25–27]. In some studies, the concentrations of saline were adjusted during the induction procedure, progressively increasing from 3% to 4% and 5% [25–27]. According to the European Respiratory Society Task Force guidelines, the standard saline solution concentration is 4.5% [14]. However, for patients at high risk of bronchoconstriction, an alternative method using isotonic saline solution should be employed (Table 1) [14]. Although hypertonic solutions are more effective than isotonic ones in inducing sputum, there is no significant difference in the cellular composition of induced sputum between isotonic and hypertonic solutions [25,26]. When conducting sputum induction, the type of nebulizer and its output should be taken into consideration. Ultrasonic nebulizers have proven to be more effective in producing adequate sputum samples compared to jet nebulizers [28]. Furthermore, a study revealed that using a high-output ultrasonic nebulizer (1.9 mL/min) and a lower output (0.7 mL/min) resulted in different cell counts and fluid-phase measures [28]. Consequently, there is a consensus in favor of using an ultrasonic nebulizer with an output of 1 mL/min to obtain a sufficient sample [14].

The duration of inhalation during sputum induction is another crucial aspect. Several studies have observed changes in cellular and biochemical elements of induced sputum during the inhalation process [29,30]. The analysis indicates that neutrophils and eosinophils predominate in sputum samples collected during the early phases of induction, while lymphocytes and macrophages become more prominent in samples collected during later phases [29,30]. This suggests that later expectorated samples originate from more peripheral airways. Changes in cellular composition during the induction process can introduce biases when performing induced sputum analysis. To ensure accurate comparisons, it is essential to maintain a constant duration of inhalation among subjects. Shorter inhalation times (15–20 min) have demonstrated similar success rates and feasibility as longer inhalation times (30 min). Most guidelines suggest an induction duration between 15 and 20 min [14]. However, it has been shown that the mean percentage of neutrophils, eosinophils, lymphocytes, and epithelial cells did not change significantly in samples obtained consecutively after 5, 10, and 15 min, as well as in the mixture of the three samples [31]. This suggests

that a 15 min induction procedure with a fixed concentration of hypertonic saline and processing of the mixed sample can be recommended for clinical settings and clinical trials.

**Table 1.** Standard and alternative methods for sputum induction. Adapted from [14].

| Standard Method | Alternative Method |
|---|---|
| Provide clear instructions to patients. | Provide clear instructions to patients. |
| ↓ | ↓ |
| Inspect all equipment and adjust ultrasonic nebulizer to approximately 1 mL/min. | Inspect all equipment and adjust ultrasonic nebulizer to approximately 1 mL/min. |
| ↓ | ↓ |
| Measure FEV1 before bronchodilator. | Measure FEV1 before bronchodilator. |
| ↓ | ↓ |
| Administer 200 µg of salbutamol via inhaler. | Administer 200 µg of salbutamol via inhaler. |
| ↓ | ↓ |
| After 10 min, measure post-bronchodilator FEV1. | After 10 min, measure post-bronchodilator FEV1. |
| ↓ | ↓ |
| Select from two options: A stable sterile saline solution at either 3% or 4.5%, or a gradually increasing saline concentration ranging from 3% to 5%. | Initial step: use 0.9% sterile saline solution and perform induction for 30 s, 1 min, and 5 min. If no sputum is produced, increase to 3% saline and induce for 30 s, 1 min, and 2 min. If unsuccessful, increase to 4.5% and induce for 30 s, 1 min, 2 min, 4 min, and 8 min. Avoid increase if isotonic saline successfully induces sputum. |
| ↓ | ↓ |
| Conduct induction in 5-min intervals for a maximum of 20 min, or at 1, 4, and 5 min with three additional 5-min periods. | Measure FEV1 after each induction interval. stop induction if FEV1 drops >20% of baseline or if symptoms occur. |
| ↓ | ↓ |
| Measure FEV1 after each induction interval. Stop induction if FEV1 drops >20% of baseline or if symptoms occur. | If the patient does not cough spontaneously, instruct them to try to cough and spit after the 4 min and 8 min periods. |
| ↓ | |
| Attempt coughing and spitting at the 10th, 15th, and 20th minutes of induction or whenever the urge arises. | |

## 4. Expectoration Technique

Various recommendations have been put forth by authors regarding the expectoration process during sputum induction, such as fasting for several hours before the procedure to prevent nausea and vomiting, rinsing the mouth with water, blowing the nose before induction, or using a nose clip [14]. However, these procedures have not shown clear benefits. Some protocols suggest interrupting the inhalation procedure at pre-defined time intervals (e.g., every 5 min) or stopping only when the patient feels the need to cough [14]. Such interruptions, however, do not affect the outcome. A particular study revealed that spitting saliva before the expectoration maneuver reduces the percentage of squamous cells in the entire sputum by 30% and increases the concentration of ECP (eosinophil cationic protein) in the supernatant by 80% [32]. Despite these observations, the impact of expectoration techniques on the feasibility and validity of the procedure remains unclear. The consensus recommends asking the patient to cough and produce sputum at the end of each set inhalation time or whenever they feel the urge to do so [14]. Additionally, some subjects,

especially those experiencing acute asthma exacerbations or COPD, can easily produce spontaneous sputum. Although spontaneously produced sputum exhibits a similar percentage of cell composition and fluid-phase mediators, it has significantly lower cell viability and poorer sample quality compared to induced sputum [33,34]. Therefore, performing induced sputum even in subjects who can spontaneously expectorate might be relevant for better comparisons with individuals who do not produce sputum spontaneously [33,34].

## 5. Sputum Processing for Cell Cytology, Immunocytochemistry, and In Situ Hybridization

Two distinct techniques for sputum processing are currently in use (Table 2) [35,36]. The first method involves selecting only the viscid portions of the sputum sample with the aid of an inverted microscope. The second technique analyzes the entire expectorate, including sputum and some saliva. To minimize salivary contamination, patients are advised to discard saliva before spitting out the sputum. The advantage of the selected plug processing is the improved quality of cytospin slides due to reduced salivary contamination [35,36]. However, this method requires more time and the use of an inverted microscope. On the other hand, processing the whole sputum is quicker, but the variable presence of saliva may dilute the sputum and affect its analysis [35,36]. Samples processed this way often contain more squamous cells, making the count of inflammatory cells challenging, particularly when squamous cells constitute more than 20% of all cells [2].

**Table 2.** Sputum processing methods for entire and selected sputum. Adapted from [14].

| ENTIRE SPUTUM | SELECTED SPUTUM |
|---|---|
| Transfer the entire sputum into a polystyrene tube previously weighed and measure its weight. | Choose a sample of sputum weighing 100–500 mg, ensuring it is saliva-free, and place it into a polypropylene tube that has been pre-weighed. |
| ↓ | ↓ |
| Add the same volume of dithioerythritol or dithiothreitol solution to the sputum. | Add dithioerythritol or dithiothreitol solution, which should be four times the weight of the selected sputum plug. |
| ↓ | ↓ |
| Use a disposable pipette to aspirate and dispense the mixture multiple times while agitating with a vortex mixer. | Using a disposable pipette, repeatedly aspirate and dispense the mixture for proper blending. |
| ↓ | ↓ |
| Place the mixture in a shaking water bath or rocker for 15 min, maintaining a temperature of either 22 °C or 37 °C. | Allow the mixture to incubate in a shaking water bath or rocker for 15 min, maintaining a temperature of either 22 or 37 °C. |
| ↓ | ↓ |
| Filter the mixture through 48 mcm nylon gauze into a conical tube that has been pre-weighed. | Filter the solution through 48 mcm nylon gauze into another pre-weighed conical tube. |
| ↓ | ↓ |
| Weigh the filtered solution in the conical tube. | Measure the weight of the filtrate in the conical tube. |
| ↓ | ↓ |
| Perform total cell count and assess the viability of the cells. | Conduct total cell count and assess the viability of the cells in the filtrate. |
| ↓ | ↓ |
| Calculate the total cell count per milliliter of the entire sputum. | Calculate the total cell count per gram of the selected sputum sample. |
| ↓ | ↓ |
| Prepare cytospins and apply Wright or Giemsa stain to the cells. | Prepare cytospins from the filtrate and apply Wright or Giemsa stain to the cells. |
| ↓ | ↓ |
| Conduct differential cell count on a minimum of 400 non-squamous cells. | Perform differential cell count on a minimum of 400 non-squamous cells present in the cytospin preparation. |

The sputum processing should be performed as soon as possible and within 2 h of collection to ensure accurate cell count and staining [37]. If there are any processing delays, the sample can be refrigerated at 4 °C for up to 8 h without affecting cell counts [38]. The use of dithiotreitol (DTT) or dithioerytriol (DTE) is recommended to break disulphide bonds in mucin molecules and release cells entrapped in mucus [39–41]. Homogenization with DTT enhances the quality of cytospins, but it may alter the levels of fluid-phase mediators. After homogenization, the suspension should be filtered through a 48 μm nylon gauze to remove mucus and debris, improving slide quality without significantly affecting the differential cell count [37,39–41].

Total cell count (TCC) and cell viability assessments are essential for sputum analysis. TCC is manually performed using a hemocytometer counting chamber, and cell viability is assessed through a Trypan blue exclusion method. Performing TCC and viability before centrifugation is recommended to avoid cell reduction [39–41]. The ideal number of cells for adequate cytospins ranges from 40 to 60 × $10^3$ [39–41]. After centrifugation, the cell pellet should be resuspended in phosphate-buffered saline solution, and cell concentration should be adjusted accordingly. Alternatively, a method using sterile minimum essential medium (MEM) has been proposed to improve cytospin quality, although it may cause some total cell loss [42].

Cytospins for the differential cell count should be stained with Giemsa or Wright's method, and at least 400 non-squamous cells (macrophages, neutrophils, eosinophils, lymphocytes, and bronchial epithelial cells) should be counted to determine the percentage of different cell types, whereas squamous cells should be reported elsewhere [43]. To monitor patient treatment accurately, monthly quality control should be performed to ensure slide reading accuracy and equipment calibration [14].

Immunocytochemistry and in situ hybridization (ISH) are less common but useful techniques for further research and diagnostics. Immunocytochemistry is best performed using the alkaline phosphatase/anti-alkaline phosphatase technique (APAAP) [9,44]. Avoiding sputum preparations with excessive mucus and squamous cells is essential to interpret immunocytochemistry results correctly. If cytokines are difficult to detect with immunocytochemistry, ISH using antisense cRNA probes can be employed. Special care must be taken to avoid RNA degradation and RNase contamination during ISH, and negative controls should be used in both techniques [9,44].

## 6. Normal Values for Cellular Composition in Induced Sputum

Understanding the normal values for the percentage and total cell counts of epithelial cells, macrophages, lymphocytes, neutrophils, and eosinophils in induced sputum is essential for interpreting these results accurately.

Epithelial cells are the guardians of the respiratory tract, forming a protective barrier against environmental irritants. In induced sputum analysis, they are a crucial component to consider. Typically, the percentage of epithelial cells in induced sputum should fall within the range of 50% to 80% [45]. This range signifies a balanced presence of these cells, reflecting a healthy respiratory epithelium. However, deviations from this range may indicate airway damage or inflammation. Macrophages are innate immune cells responsible for clearing foreign particles and microbes from the airways. In the context of induced sputum analysis, macrophage percentage and total cell counts are essential markers of airway health. Normally, macrophages should constitute around 50% to 80% of the total cell count [45]. These values suggest efficient clearance mechanisms and a well-functioning immune response in the airways. Lymphocytes are adaptive immune cells that play a significant role in chronic inflammatory conditions and infections. A typical range for lymphocyte percentage in induced sputum is 10% to 25% [45]. Elevated levels of lymphocytes may indicate an ongoing immune response, such as in the case of viral infections or autoimmune disorders. Neutrophils are the first responders to infections and are often seen in higher numbers during acute inflammation. In induced sputum analysis, the normal range for neutrophil percentage is approximately 20% to 50% [45]. Elevated

neutrophil counts can signal an acute inflammatory process, such as a bacterial infection or exacerbation of chronic respiratory diseases like COPD. Eosinophils are specialized white blood cells involved in allergic and parasitic responses. In the context of induced sputum analysis, eosinophil levels are especially critical in diagnosing and managing allergic airway diseases like asthma. The normal range for eosinophil percentage typically lies between 0% and 5% [45]. Elevated eosinophil counts, often above 2%, are indicative of eosinophilic airway inflammation and can guide treatment decisions, such as corticosteroid therapy. While these ranges serve as general guidelines for interpreting induced sputum results, it is essential to remember that individual variability exists. Furthermore, these values may shift depending on factors such as age, smoking status, and underlying health conditions [45]. Interpretation should always consider the broader clinical context. Deviations from these normal values can provide valuable insights for healthcare professionals. For instance, an increase in neutrophil percentage and total cell count might suggest a bacterial infection, while a rise in eosinophils could indicate an allergic response. These findings guide clinicians in making accurate diagnoses and tailoring treatment plans to address the specific underlying cause of respiratory symptoms.

## 7. Conclusions

Compared to other airway cell collection techniques like biopsies and bronchoalveolar lavages, induced sputum has several advantages: it is simple, well tolerated, safe, reproducible, cost-effective, and non-invasive. This makes it suitable for large-scale and repeated sampling over time, offering a preferred alternative for airway sampling. Although broncho-absorption allows for the collection of mucosal lining fluid and higher mediator concentrations, it requires bronchoscopy and is more invasive than induced sputum.

Despite its benefits, the induced sputum technique does have limitations. It requires medical supervision, thorough patient instructions, and patient cooperation and condition. The feasibility in children older than 6 years has been reported, but it may be difficult in younger children. Additionally, the technique is currently restricted to research services and specialized centers due to its technical demands and the need for trained staff. The success rate of sputum induction and analysis is generally around 80%, and caution should be exercised when comparing patient cohorts, taking age, gender, and smoking status into account.

**Author Contributions:** Conceptualization, S.D. and A.B.; methodology, A.C.; software, A.C.; validation, S.S.; formal analysis, A.C.; investigation, S.D.; resources, S.D.; data curation, S.D.; writing—original draft preparation, S.D.; writing—review and editing, G.E.C.; visualization, S.S.; supervision, G.E.C.; project administration, A.B.; funding acquisition, S.D. All authors have read and agreed to the published version of the manuscript.

**Funding:** This research received no external funding.

**Institutional Review Board Statement:** Not applicable.

**Informed Consent Statement:** Not applicable.

**Data Availability Statement:** No new data were created for this review.

**Conflicts of Interest:** The authors declare no conflict of interest.

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
