# Peer review of "Methodological Aspects of Induced Sputum"

_arm, doi:10.3390/arm91050031_

Round 1

Reviewer 1 Report

The manuscript is well written as an instruction manual for clinicans. There are no new data in the manuscript. 

The only comments I can give are to look through figure 3 and 4 regading hypenation and and abbreviation.s (avoid if possible in figures)

Author Response

Reviewer 1

The manuscript is well written as an instruction manual for clinicans. There are no new data in the manuscript. 

The only comments I can give are to look through figure 3 and 4 regading hypenation and and abbreviation.s (avoid if possible in figures)

R- Thanks for your evaluation and for your comment. We have modified figure 3 and 4 based on your suggestions. 

Reviewer 2 Report

Dear authors,

First of all thank you for your interesting review in a topic where information lacks. I have however some questions for you to address:

L30: what type of review did you perform? This should be detailed in the abstract

L38: in patients with mild/moderate it was expected to be safe – the phrase even in would apply for example for patients with severe disease

L50: Do you have patient consent for this image? This should be included in the legend

L104 and L119: please reference all this information

L242-L256: This information is only included in the conclusion. It should be detailed in the rest of the paper also

Author Response

Dear authors,

First of all thank you for your interesting review in a topic where information lacks. I have however some questions for you to address:

L30: what type of review did you perform? This should be detailed in the abstract

L38: in patients with mild/moderate it was expected to be safe – the phrase even in would apply for example for patients with severe disease

L50: Do you have patient consent for this image? This should be included in the legend

L104 and L119: please reference all this information

L242-L256: This information is only included in the conclusion. It should be detailed in the rest of the paper also

R- Dear Reviewer, many thanks for your very constructive comments which helped us to improve our manuscript. We have addressed all your questions. Please see our new version of the manuscript.

Reviewer 3 Report

The manuscript „Methodological aspects of induced sputum” is a well written systematic review of latest knowledge of performing, handling, and evaluation of induced sputum. Although this review lack novelty, but puts together very practical and useful information, and will be helpful for medical stuff and specialists because contains many practical instructions. I have some minor remarks:

1.       The authors should put it clear that after 20% drop of FEV1 the procedure should be immediately and irretrievably stopped.

2.       The description of most frequently used range of norm for percentage and total cell counts of epithelial cells, macrophages, lymphocytes, neutrophils, eosinophils in induced sputum should be added to manuscript.

Author Response

The manuscript „Methodological aspects of induced sputum” is a well written systematic review of latest knowledge of performing, handling, and evaluation of induced sputum. Although this review lack novelty, but puts together very practical and useful information, and will be helpful for medical stuff and specialists because contains many practical instructions. I have some minor remarks:

  1. The authors should put it clear that after 20% drop of FEV1 the procedure should be immediately and irretrievably stopped.

R- We agree with the reviewer. We have modified that sentence accordingly.

  1. The description of most frequently used range of norm for percentage and total cell counts of epithelial cells, macrophages, lymphocytes, neutrophils, eosinophils in induced sputum should be added to manuscript.

R- Thanks for your suggestion. We have now included a new chapter about normal values in induced sputum (please see chapter 6).

Round 2

Reviewer 2 Report

Information from L84 to L95 should be referenced.

Author Response

Thank you for the comment. We provided the requested references.